# Depletion of CD8αβ^+^ T Cells in Chickens Demonstrates Their Involvement in Protective Immunity towards Marek’s Disease with Respect to Tumor Incidence and Vaccinal Protection

**DOI:** 10.3390/vaccines8040557

**Published:** 2020-09-24

**Authors:** Supawadee Umthong, John R. Dunn, Hans H. Cheng

**Affiliations:** 1Microbiology and Molecular Genetics Program, Michigan State University, East Lansing, MI 48823, USA; umthongs@msu.edu; 2USDA, ARS, US National Poultry Research Center, Avian Disease and Oncology Laboratory, East Lansing, MI 48823, USA; john.dunn@usda.gov

**Keywords:** Marek’s disease, vaccine, CD8+ T cells, tumor

## Abstract

Marek’s disease (MD) is a lymphoproliferative disease in chickens caused by Marek’s disease virus (MDV), a highly oncogenic alphaherpesvirus. Since 1970, MD has been controlled through widespread vaccination of commercial flocks. However, repeated and unpredictable MD outbreaks continue to occur in vaccinated flocks, indicating the need for a better understanding of MDV pathogenesis to guide improved or alternative control measures. As MDV is an intracellular pathogen that infects and transforms CD4+ T cells, the host cell-mediated immune response is considered to be vital for controlling MDV replication and tumor formation. In this study, we addressed the role of CD8+ T cells in vaccinal protection by widely-used monovalent (SB-1 and HVT) and bivalent (SB-1+HVT) MD vaccines. We established a method to deplete CD8+ T cells in chickens and found that their depletion through injection of anti-CD8 monoclonal antibodies (mAb) increased tumor induction and MD pathology, and reduced vaccinal protection to MD, which supports the important role of CD8+ T cells for both MD and vaccinal protection.

## 1. Introduction

The cytotoxic T-lymphocyte (CTL) response generated from activated CD8+ T cells is an effective immune response that combats infectious pathogens [1]. For example, studies have provided evidence that the CD8+ T cell response is associated with controlling (1) infections by intracellular bacteria, such as *Mycobacterium tuberculosis* [2,3,4,5], *Salmonella typhimurium* [6,7], and *Listeria monocytogenes* [8], (2) infections by intracellular viruses, such as herpes simplex virus type 1 [9,10,11], hepatitis B virus [12,13], and respiratory syncytial virus [14,15], and (3) intracellular infections by protozoa such as *Plasmodium* [16,17,18,19], *Toxoplasma gondii* [20], and *Trypanosoma cruzi* [21]. The CD8+ CTL response functions by recognizing foreign antigens presented by MHC class I of infected cells, then destroying the cells by releasing perforin and granzyme to induce apoptosis. CD8+ T cells also produce and secrete cytokines, such as IFN-γ that directly inhibits viral replication, induces maturation of macrophages, and activates the natural killer (NK) cell response. CD8+ effector cells can also secrete chemokines to further prolong the CD8 T cell response. In herpesvirus infections, activation of the CD8+ CTL response is generally developed against viral glycoproteins, which are part of the viral envelope [22,23,24,25].

Marek’s disease (MD) is a CD4+ T cell lymphoma in chickens caused by the infection of an alphaherpesvirus called Marek’s disease virus (MDV, aka *Gallid alphaherpesvirus 2*). MDV is a highly cell-associated virus as the spread of infection within the host is via cell-to-cell contact [26]. Consequently, although maternal antibody has been shown to reduce the severity of MD [27,28], the cell-mediated immune response is believed to be more important than the humoral immune response. CD8+ CTL has been reported to respond to MDV glycoproteins [25]. Furthermore, Omar and Schat [29] reported that SB-1 vaccine could induce a CD8+ CTL response. These results emphasize the importance of the CD8+ CTL response in protection against MDV and in vaccine-induced protective immunity.

Several vaccines have been developed from attenuated serotype 1 (e.g., CVI988), serotype 2 (e.g., SB-1), and serotype 3 turkey herpesvirus (e.g., HVT strain FC126) viruses to protect chickens from MD. SB-1 and HVT strain FC126 are commonly used commercial MD vaccines. These two vaccines can be used alone as monovalent vaccines or in combination as a bivalent vaccine, with the latter demonstrating greater protection compared to either monovalent vaccine alone. However, no MD vaccine can prevent MDV infection, clear the pathogenic MDV, or prevent the shedding of pathogenic MDV from the host.

In this study, we investigated the effect of CD8+ T cells on both MD incidence and vaccinal protection. We optimized a method to specifically deplete CD8+ T cells in vivo by injecting antibody purified from the culture supernatant of LC-4 hybridoma cells, which secrete anti-chicken CD8. We monitored CD8+ T cell populations after immunization in normal vs. depleted CD8 T cell chickens. We finally measured the protection, MD pathology (tumor and nerve enlargement incidence) after vaccination in control chickens compared to in CD8-depleted chickens. From this study, we provided evidence that CD8+ T cells are important for MD protection and prevention in both unvaccinated as well as vaccinated chickens.

## 2. Materials and Methods

### 2.1. LC-4 Hybridoma Culture

LC-4 hybridoma cells, which secrete anti-chicken CD8 antibody [30,31], were used as a source to produce the antibody for in vivo CD8 depletion. The hybridoma cells were initially maintained in Dulbecco’s Modified Eagle’s Media (DMEM) with 20% fetal bovine serum (FBS) (Thermo Fisher Scientific, Waltham, MA, USA) in a 37 °C incubator with 5% CO_2_ for three days. Later, the cells were passed and maintained in the sequentially lower concentration of FBS at 10%, 5%, and 2% for every three days. Afterwards, the cells were passed and maintained in protein-free hybridoma medium II (PFHMII) supplied with 0.2% chemically defined lipid concentrate (Life Technologies, Carlsbad, CA, USA). The CELLine 1000 bioreactor flask for high-density suspension cells was used (Wheaton, IL, USA) to grow the hybridoma cells for large-scale antibody production following recommendations of the company.

### 2.2. Purification and Characterization of Anti-Chicken CD8 mAb

The LC-4 hybridoma culture supernatant was collected every 3–4 days depending on the density of the cells. Proteins in the culture supernatant were precipitated by the addition of saturated ammonium sulfate solution (Thermo Fisher Scientific) until the solution became turbid. The solution was centrifuged at 3000× *g* for 30 min at 4 °C. The pellet was resuspended in a small volume of phosphate buffer saline (PBS) and dialyzed against PBS overnight at 4 °C using Slide-A-Lyzer 10K dialysis cassettes (Thermo Fisher Scientific). The resulting antibody was measured for concentration using a Nanodrop 8000 spectrophotometer (Thermo Fisher Scientific) and 2 μL and 10 μL of purified mAb was further analyzed for purity by Coomassie staining and Western blot. For the latter, the blot was probed with 1:2000 of goat anti-mouse IgG horseradish peroxidase (HRP), washed, and visualized using the Pierce^TM^ ECL Western Blotting Substrate (Thermo Fisher Scientific).

### 2.3. Characterization of Anti-CD8 mAb Binding Activity

Anti-chicken CD8 antibody was tested for its specific capacity to bind to chicken CD8+ T cells by immunohistochemistry (IHC) and flow cytometry. For IHC, splenic and thymic samples from 5-day old chickens were cut, embedded in optimum cutting temperature (OCT) compound (Sakura Finetek, Torrance, CA, USA), and kept at −80 °C until use. The tissue was cut using a cryostat microtome HM505 Microm and fixed on tissue slides with acetone. The fixed tissues were stained with 1:100 purified anti-chicken CD8 mAb and IHC staining was performed using VECTASTAIN Elite ABC HRP mouse IgG and ImmPACT DAB HRP substrate kits (Vector Laboratory, Burlingame, CA, USA). The stained sections were observed under a microscope. For flow cytometry, peripheral blood mononuclear cells (PBMCs) were purified from whole blood of one of the chickens using Histoplaque-1077 (Sigma Aldrich, St. Louis, MO, USA). The PBMCs were stained with purified 1:100 anti-chicken CD8 mAb, washed, and stained with Alexa Fluor 488 goat anti-mouse IgG (H+L) (Life Technologies), and detected using flow cytometry (BD FACS Calibur^TM^, BD Biosciences, San Jose, CA, USA).

### 2.4. Birds

All chickens were United States Department of Agriculture (USDA), Agricultural Research Service (ARS), Avian Disease and Oncology Laboratory (ADOL) line 15I_5_ × 7_1_ white leghorns, a F_1_ hybrid cross of MD susceptible line 15I_5_ males and line 7_1_ females. Chicks were from maternal antibody positive (ab+) hens, which had been vaccinated with 1000 plaque forming units (PFU) each of HVT and SB1 at hatch followed by CVI988/Rispens at 25 weeks of age for exposure to all three serotypes. All bird experiments were approved by the ADOL Institutional Animal Care and Use Committee; approval no. 2018-06.

### 2.5. Viruses

SB-1 and HVT strain FC126 were grown in duck embryo fibroblast (DEF) cells and used as vaccines, while MDV serotype 1 strain Md5 (passage 8) was produced in chicken embryonic fibroblasts (CEF) and used for challenge. The titers of the virus were determined by counting the plaque numbers of each virus upon infection in either CEF or DEF cells. The titers were adjusted to the corresponding PFU needed before vaccination or challenge.

### 2.6. Optimization of Route and Dosage for CD8 Depletion In Vivo

The protocol for antibody depletion was optimized to compare routes of injection (intra-peritoneal [IP] vs. intra-venous [IV] injection) and dosage of injections (1 or 3 mg of anti-chicken CD8 mAb). Newly hatched chicks were randomly divided into five groups each with five birds per group: (1) control (no injection), (2) 1 mg anti-chicken CD8 mAb by IP, (3) 3 mg anti-chicken CD8 mAb by IP, (4) 1 mg anti-chicken CD8 mAb by IV, and (5) 3 mg anti-chicken CD8 mAb by IV. Blood was collected at 1 day post-challenge (DPC), and 1, 2, 3 and 4 weeks post challenge (WPC). PBMCs were separated using Histopaque (Sigma-Aldrich, St. Louis, MO, USA) and stained for CD8αβ+ T cells using mouse anti-chicken CD8α-FITC and mouse anti-chicken CD8β-PE (Southern Biotech, Birmingham, AL, USA) and analyzed by flow cytometry (BD FACSCalibur, BD Biosciences).

### 2.7. Determination of CD3^+^CD4^+^ T Cells and CD8αβ^+^ T Cells

Spleens were collected from five birds per group from unvaccinated, SB-1 vaccinated, HVT vaccinated, and bivalent vaccinated groups; both untreated and CD8-depleted chickens. Samples were homogenized and stained for CD3^+^CD4^+^ and CD8αβ^+^ T cell populations. Anti-chicken CD3 Alexa Fluor 647, CD4 PE, CD8α FITC, CD8β-SPRD monoclonal antibodies (Southern Biotech) were used for multi-color flow cytometry detection. The results were analyzed by FlowJo software version 10.0.8 (Ashland, OR, USA).

### 2.8. Measurement of Vaccinal Protection

The efficacy of MD vaccinal protection was tested in vivo. Newly hatched chickens were divided into ten groups as described in Table 1. Chickens were vaccinated with 2000 PFU of monovalent SB-1, 2000 PFU of monovalent HVT, or bivalent SB-1+HVT (1000 PFU each) on the first day of age. For CD8 depletion, 1 mg (100 μL) of anti-chicken CD8 mAb was administered by IV injection via jugular vein using a BD ultra-fine insulin syringe 31G needle on days 2, 3, and 4 of age. Chickens were then challenged with 1000 PFU of pathogenic MDV. Anti-chicken CD8 mAb was injected weekly with the same dose and same route after challenge until the end of the study (eight weeks) (Figure 1). Percent protection of the vaccine was calculated based on the formula below:$$\%\;\mathrm{Protection}\;=\;\frac{\left(\%\mathrm{MD}\;\mathrm{in}\;\mathrm{unvaccinated}\;\mathrm{group}\;-\;\%\mathrm{MD}\;\mathrm{in}\;\mathrm{vaccinated}\;\mathrm{group}\right)\;\times \;100\;}{\%\mathrm{MD}\;\mathrm{in}\;\mathrm{unvaccinated}\;\mathrm{group}\;}$$

Percent tumor induction was measured based on the number of chickens that developed tumors in each vaccination group compared to the control group. A positive case of MD required nerve (vagus, sciatic, and/or brachial) enlargements and/or tumor development. All observed pathology was counted and scored from 1 (low severity) to 4 (high severity) for nerve enlargement and lymphoid atrophy.

## 3. Results

### 3.1. Production of Anti-Chicken CD8 mAb by Culturing LC-4 Hybridoma Cells

To avoid using animals, anti-chicken CD8 mAb was produced by culturing LC-4 hybridoma cells in CELLine 1000 bioreactor flasks, which was able to yield up to 100 mg of the antibody per week. We checked the purity of the antibody by Coomassie staining as shown in Figure 2A. The heavy chain (~50 kDa) and the light chain (~25 kDa) of mouse IgG antibody was confirmed by Western blot as shown in Figure 2B.

### 3.2. Binding Activity of Anti-CD8 mAb to Chicken CD8 T Cells

Prior to our in vivo studies, the binding activity of anti-CD8 mAb was confirmed by IHC staining for CD8+ T cells in splenic and thymic sections, and by flow cytometry staining for CD8+ T cells in chicken PBMCs. The results confirmed that the anti-chicken CD8 mAb specifically bound CD8 T cells in the spleen (Figure 2D) and in thymus (Figure 2F) compared to the controls (Figure 2C,E, respectively). Anti-chicken CD8 mAb also showed binding activity to CD8+ T cells in total PBMCs obtained from blood observed by flow cytometry (Figure 2G).

### 3.3. IV Injection is the Better Route of Injection to Deplete Chicken CD8+ T Cells Using Anti-Chicken CD8 mAb

After confirming the anti-CD8 mAb-binding efficacy, the purified antibody was then used to inject chickens to specifically deplete the CD8 T cell population. We tested the protocol to deplete CD8+ T cells by varying both the dose and route of injection. Our results indicated that 1 mg injection via IV route was feasible and the most effective way to reduce the level of CD8+ T cells, and the reduction remained up to 4 weeks post injection (WPI) as shown in Figure 3. All chickens in the CD8-depleted group were healthy and developed no symptoms or signs of sickness after CD8 depletion. Thus, 1 mg of antibody administered by IV injection was used for further experiments.

### 3.4. Proportion of CD4+ and CD8αβ+ T Cells in Normal and CD8 Depleted Chickens after Vaccinations

We measured levels of CD4+ cells to confirm that the anti-chicken CD8 mAb specifically depletes only CD8αβ+ T cells and not CD4+ T cells in live birds. Total splenocytes were collected and stained for the respective T cell markers. The level of CD4+ and CD8αβ+ T cells were monitored 1-day DPC until 7 WPC. The results showed that levels of CD4+ T cells were not changed by the injection of anti-chicken CD8 mAb in all vaccinations (Figure 4A). Injection of anti-chicken CD8 mAb decreased the level of CD8αβ+ T cells 2–4 times compared to control without antibody injection and this happened in all treatments including unvaccinated control, SB-1, HVT, and bivalent SB-1+HVT vaccination. However, the levels of CD8αβ+ T cells can be depleted only up to 4 weeks as the levels were about the same as control after 4 WPC (Figure 4B).

### 3.5. CD8+ T Cells Play an Important Role for MD Resistant and Vaccinal Protection

We investigated whether the CD8+ T cell response contributes to MD resistance, in general, and vaccinal protection induced by SB-1, HVT, and SB-1+HVT vaccines. The results showed protection in the SB-1, HVT, or SB-1+HVT vaccinated groups both with and without CD8-T cell depletion (Figure 5 and Table 1). However, depletion of CD8+ cells reduced the protective level of SB-1 and HVT monovalent vaccines as well as that of bivalent vaccine (Figure 5A). In addition, depletion of CD8+ T cells increased tumor incidence in unvaccinated and the monovalent vaccinated group but none of the chickens in the bivalent vaccinated group developed tumors in either the control CD8 or depleted-CD8 groups (Figure 5B). For MD pathology observed from nerve lesions, higher sciatic cumulative lesion scores (greater enlargement) were observed when CD8 T cells were depleted in all treatments including the unvaccinated-challenged control and all vaccinations (Figure 5C). Interestingly, brachial and vagal nerve lesions remained the same in the unvaccinated group when CD8 cells were depleted but the lesions were increased in vaccinated groups especially in those with HVT and SB-1+HVT vaccination (Figure 5D,E). Our results show that depletion of CD8 T cells increased nerve lesions after vaccinal protection by HVT and bivalent (SB-1+HVT). These results indicate that CD8+ T cells are involved in protection against MD infection and also involved in vaccinal protection by SB-1 and HVT vaccines.

## 4. Discussion

The CD8+ CTL response is considered as an effective immune response against intracellular pathogens and cancer cells. Given the cell-associated and oncogenic nature of MDV, the CD8+ T cell response should be necessary for MD protection. Several studies have shown the involvement of CD8 T cells in MD. For instances, Omar and Schat showed the induction of CD8 T cells that target MDV antigens such as gB, Meq, pp38, and ICP4 [32], while Markowski-Grimsrud and Schat provided evidence of a CTL response against late MDV glycoproteins [25,33]. In addition, transcriptome analysis of host response during MDV infection also showed induction of a CTL response, e.g., up-regulation of granzyme A homolog-gene compared to uninfected chickens [34]. Prior studies on MD vaccines showed that CD8+ T cells were induced upon vaccinations and the induction of CD8+ T cells correlated with vaccinal protection [35,36,37]. However, these studies utilized thymectomy to deplete the number of CD8+ T cells, which compared to bursectomy, is difficult given that there are seven lobes on each side of the neck in hard to access regions, especially in small chicks. Furthermore, thymectomy removes both CD4+ and CD8+ T cells. In this study, we optimized an alternative and durable way to specifically deplete CD8+ T cells in vivo by using antibody injection and then investigated whether the CD8+ T cell response was involved in both MD resistance and vaccinal protection.

First, in order to minimize the use of animals, we produced anti-chicken CD8 mAb by large-scale culturing of LC-4 hybridoma cells in vitro instead of producing antibody in mice through the ascites induction method [38]. LC-4 hybridoma cells secrete monoclonal antibody against chicken CD8+ T cells and the antibody produced specifically binds to chicken CD8+ T cells found in the spleen, thymus, and PBMCs. The optimization of route of injection indicated that IV injection was more effective than IP injection and we found that, in general, 1 mg via IV injection was sufficient.

The levels of CD4+ and CD8αβ+ T cells were measured in all vaccinated groups, and it was found that the levels of CD4+ T cells were not different in control vs. CD8-depleted birds ensuring that anti-CD8 injection did not interfere with the level of CD4+ T cells. Our finding agrees with the results of Kondo et al. [30]. However, we observed that the level of CD4+ T cells were slightly higher in CD8+ T cells of the depleted, unvaccinated-challenged group compared to normal chickens, after 4 weeks post challenge. This could be due to higher lymphoma induction in the CD8-depleted group implying that CD8+ T cells are involved in controlling lymphoma development in MDV infected chickens. As observed by the reduction of CD8+ T cells after challenge, antibody injection depleted CD8+ T cells up to around 4 WPC. This result can be explained in several ways. First, as the chickens became bigger, injection of 1 mg of antibody after 4 WPC was not sufficient to deplete all CD8+ T cells. Second, all chickens were challenged and some were vaccinated. This may cause the strong induction of CD8+ T cells to the level that the antibody injection was not sufficient to deplete the high number of CD8+ T cells after challenge.

Percent MD development is based up on number of chickens that develop MD, which includes nerve enlargement and tumor. Reduction of percent protection in unvaccinated group, monovalent SB-1 vaccinated group, and bivalent vaccinated group was observed in CD8-depleted chickens indicating that CD8+ T cells are involved in MD resistance from both infection and vaccinations.

Higher number of nerve lesions as well as percent tumor induction in the unvaccinated group and monovalent vaccinated group were observed when CD8+ T cells were depleted. Thus, CD8+ T cells seem to be required for both anti-viral and anti-tumor response. However, we did not observe higher tumor incidence in the bivalent vaccinated group after CD8 were depleted. This suggests that SB-1+HVT bivalent vaccine may also promote a strong anti-tumor response and the response likely requires fewer CD8+ T cells. Rather, depletion of CD8+ T cells plays a role in the anti-viral response in bivalent vaccination as shown by the reduction of protection (due to some chickens started to show MD incidence observed by nerve enlargement after CD8+ T cells were depleted). Our bivalent vaccination data agrees with work performed by Morimura using CVI988 as a vaccine model injected into CD8 depleted chickens. Their study showed that CVI988 vaccine involved in prevention of MD infection but was not essential to prevent lymphoma formation [35]. Another reason could be due to the ability of CVI988 and bivalent SB-1+HVT to reduce pathogenic MDV replication at the first place resulting in the less likely chance of tumor outcome. In our study, we did not include CVI988 in our experiment as our major goal was to understand whether the CD8 CTL response is likely the reason for protective synergy provided by a bivalent SB-1+HVT vaccine compared to monovalent of single SB-1 or HVT [39]. In addition, Witter’s study in 1987 showed no synergism observed when CVI988 was administered together with SB-1 or HVT FC126 [40]. Based on our findings, CD8+ T cells are important for not only bivalent SB-1+HVT vaccine but also monovalent SB-1 and monovalent HVT vaccines. Thus, the protective response provided by CD8 may not be the only reason for the observed protective synergy of SB-1+HVT bivalent vaccines.

## 5. Conclusions

In summary, our study provides the optimized and durable approach to study the effect of CD8 T cell in MD resistance and MD protection by vaccines. In addition, we provide evidence of the vital role that CD8+ T cells play in MD prevention with respect to anti-viral and anti-tumor responses in both unvaccinated and MD vaccinated birds.

## Figures and Tables

**Figure 1 vaccines-08-00557-f001:**
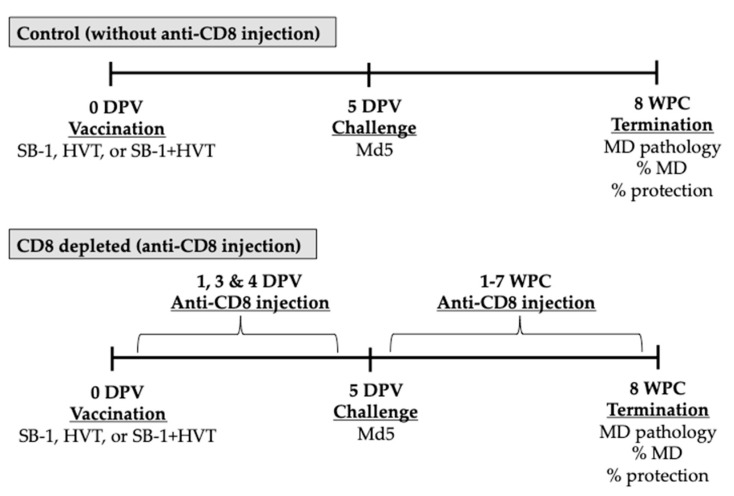
Schematic of experiments performed to test the effect of CD8 T cell on MD and vaccinal protection. Chicks were vaccinated with SB-1, HVT, or bivalent SB-1+HVT vaccine on the day of hatch, and later injected with PBS only or purified 1 mg of anti-CD8 mAB on 1, 3, and 4 days post-vaccination (DPV). Chickens were challenged with Md5 pathogenic MDV 5 DPV. One mg of anti-CD8 mAb was administered weekly to deplete chicken CD8 T cells until 8 weeks post challenge. Percent MD was determined based on MD pathology including nerve lesions and tumor development.

**Figure 2 vaccines-08-00557-f002:**
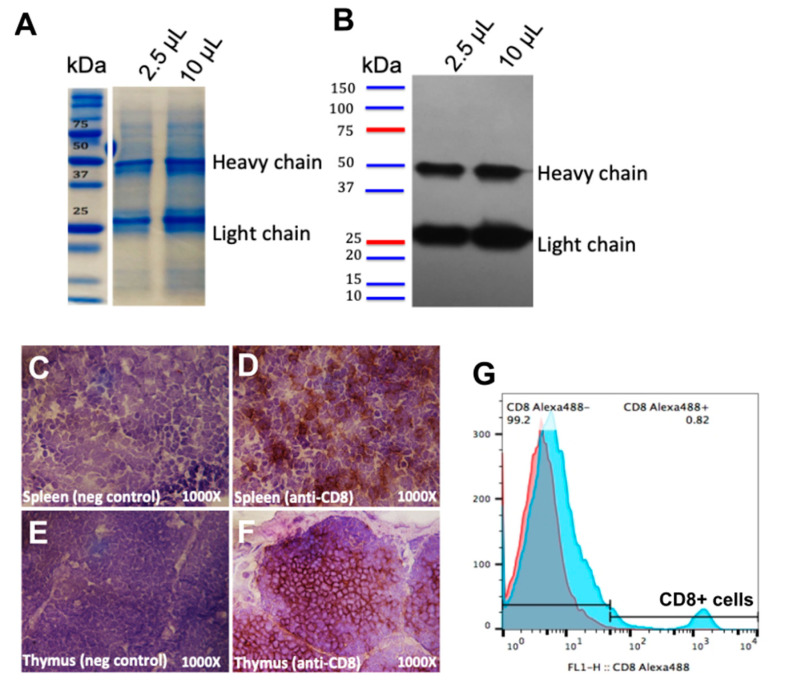
Purification and binding confirmation of anti-CD8 mAb. 2.5 μL or 10 μL of purified mAb collected from LC-4 hybridoma cells were run on SDS-PAGE and the purity of the protein is shown by Coomassie staining (**A**). Heavy and light chain IgG of the anti-CD8 antibody were detected by Western blotting using goat anti-mouse IgG HRP (**B**). The binding of the CD8 mAb was observed in chicken CD8+ T cells in spleen (**C**,**D**), and thymus (**E**,**F**), using IHC; control (staining using a secondary antibody only) (**C**,**E**), or with purified anti-CD8 mAb (**D**,**F**). The binding of anti-CD8 mAb was also confirmed in chicken PBMCs by cell surface staining using anti-CD8 mAb as a primary antibody and goat anti-mouse Alexa Fluor 488 as a secondary antibody to detect CD8+ T cells by flow cytometry (**G**).

**Figure 3 vaccines-08-00557-f003:**
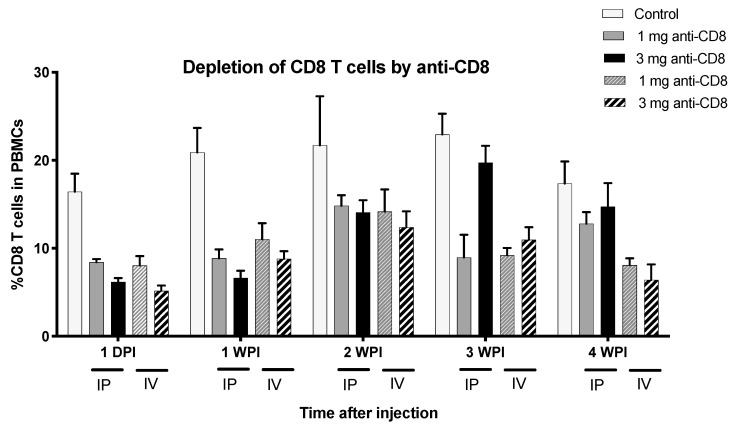
Depletion of CD8+ T cells by antibody injection in vivo. PBMC were extracted from whole blood from normal control chickens or anti-CD8 injected chickens and flow cytometry was used to determine percent of CD8αβ+ T cells in PBMC. Each bar of each color represents percent of CD8αβ+ T cells in PBMC in control, 1 mg anti-CD8 IP, 3 mg anti-CD8 IP, 1 mg anti-CD8 IV, and 3 mg anti-CD8 IV injections, respectively. DPI = days post injection.

**Figure 4 vaccines-08-00557-f004:**
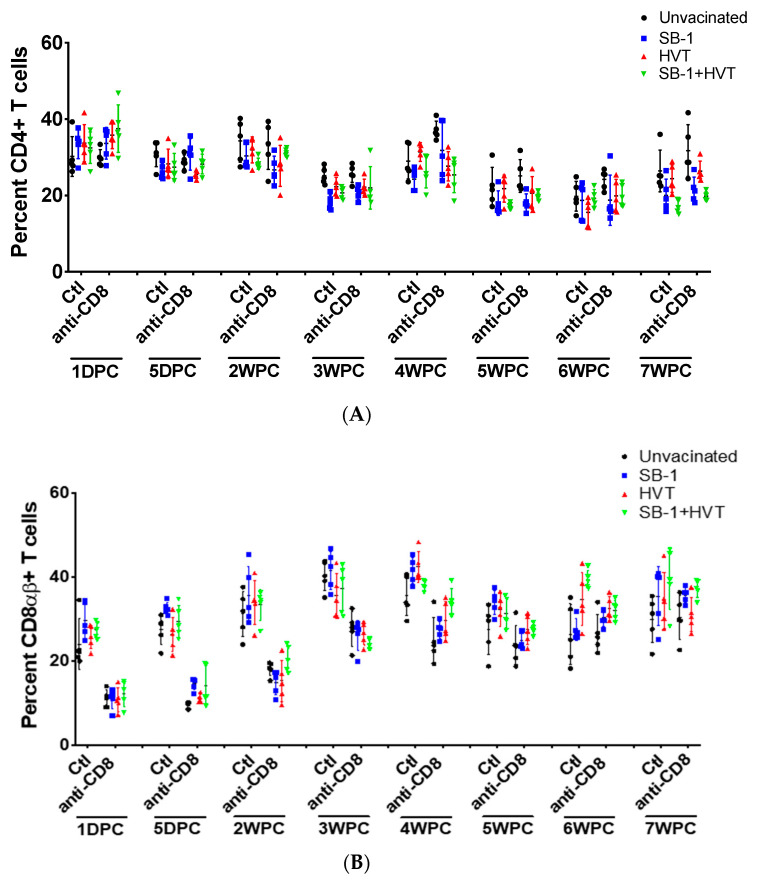
Levels of CD4+ and CD8αβ+ T cells after anti-chicken CD8 mAb injection. Levels of CD4+ (**A**) and CD8αβ+ T cells (**B**) in spleen of control (Ctl) and CD8-depleted (anti-CD8) groups in all types of vaccination were observed through the duration of the study from 1 DPC to before termination at 7 WPC, using flow cytometry.

**Figure 5 vaccines-08-00557-f005:**
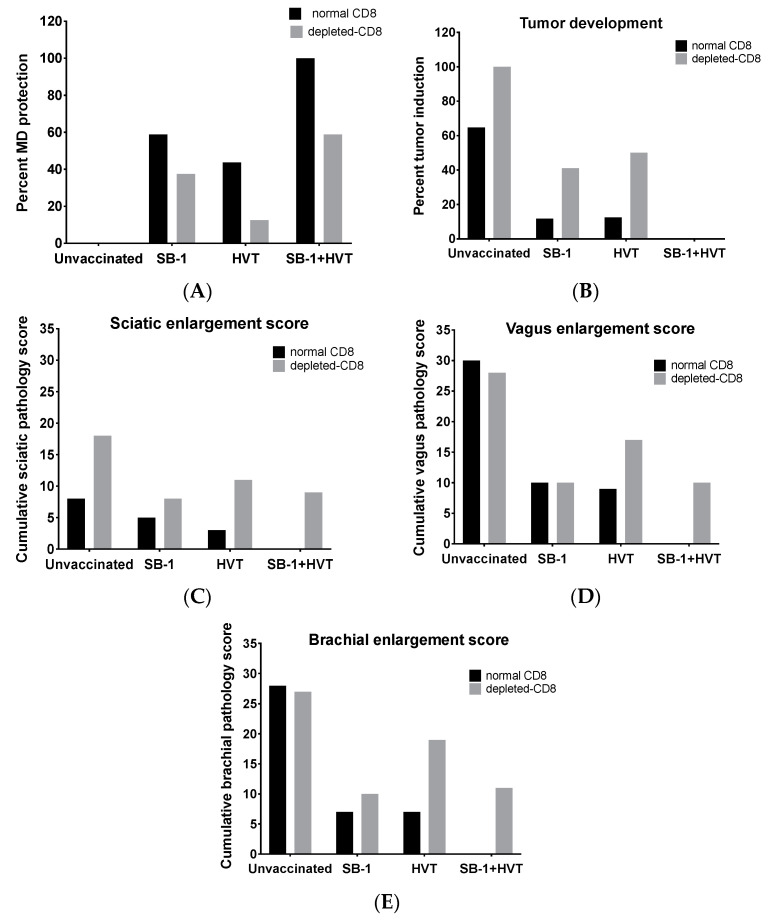
Effect of CD8+ T cells on MD pathology and MD vaccine protective efficacy. Control chickens or CD8-depleted chickens were immunized with SB-1, HVT, or SB-1+HVT vaccines, and then challenged with pathogenic MDV strain MD5 and kept for 8 weeks. Percent protection (**A**), percent tumor development (**B**), and nerve lesions including sciatic enlargement (**C**), vagus enlargement (**D**), and brachial enlargement (**E**) were determined during necropsy.

**Table 1 vaccines-08-00557-t001:** Experimental scheme to measure the effect of CD8 T cell on protection of SB-1, HVT, and SB-1+HVT vaccines.

Group	Vaccine	CD8 Depletion	Challenge (5 DPV)	Number of Chickens
1	Unvaccinated	No	No	17
2	Unvaccinated	No	1000 PFU of Md5	17
3	SB-1	No		17
4	HVT	No		17
5	SB-1+HVT	No		17
6	Unvaccinated	Yes	No	17
7	Unvaccinated	Yes	1000 PFU of Md5	17
8	SB-1	Yes		17
9	HVT	Yes		17
10	SB-1+HVT	Yes		17
**Total number of chickens**				170

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
