# Peer review of "Depletion of CD8αβ+ T Cells in Chickens Demonstrates Their Involvement in Protective Immunity towards Marek’s Disease with Respect to Tumor Incidence and Vaccinal Protection"

_vaccines, 2020, doi:10.3390/vaccines8040557_

Round 1

Reviewer 1 Report

In this study, the authors addressed the role of CD8+T cells in vaccinal protection by widely-used monovalent (SB-1 and HVT) and bivalent (SB-1+HVT) Marek’s disease (MD) vaccines. They hypothesized that depletion of CD8+ T cells would increase tumor incidence in unvaccinated birds and reduce the protective efficacy in MD vaccinated chickens. They found that depletion of CD8+ T cells through injection of anti-CD8 antibodies increased tumor induction and MD pathology, and reduced vaccinal protection to MD.

The paper can be accepted for the publication after some minor revisions.

Comments:

  1. Define all acronyms used in the manuscript.
  2. Compare your results obtained in this study with other existing in the literature.
  3. There are some typos. The authors should carefully read the manuscript.
  4. In this study, why did you only consider the CTL immune response?
    Does humoral immunity have no effect?
  5. Check the references used in this study.

Author Response

We greatly appreciate the prompt and thorough comments that we made by both reviewers. We have addressed each comment which is highlighted in yellow, as well as made changes in the manuscript that are also highlighted in yellow. We hope that these modifications make the manuscript more valuable to the community.

Reviewer #1: In this study, the authors addressed the role of CD8 T cells in vaccinal protection by widely-used monovalent (SB-1 and HVT) and bivalent (SB-1+HVT) Marek’s disease (MD) vaccines. They hypothesized that depletion of CD8+ T cells would increase tumor incidence in unvaccinated birds and reduce the protective efficacy in MD vaccinated chickens. They found that depletion of CD8 T cells through injection of anti-CD8 antibodies increased tumor induction and MD pathology, and reduced vaccinal protection to MD. The paper can be accepted for the publication after some minor revisions.

Comments:

1. Define all acronyms used in the manuscript.

As stated in the Instructors for Authors, all abbreviations have been defined the “first time they appear in the abstract, main text, and in figure or table captions.”

2. Compare your results obtained in this study with other existing in the literature.

We have added a few more references (35, 37, and 38). Hopefully these additions plus the original ones provide a good summary of the field.

3. There are some typos. The authors should carefully read the manuscript.

We apologize for not catching all the errors. We have gone through the manuscript very carefully and have highlighted changes that we found. We hope that the manuscript is satisfactory now.

4. In this study, why did you only consider the CTL immune response? Does humoral immunity have no effect?

As written on lines 42-44, the primary reason for pursing the CTL immune response is that MDV is a highly cell-associated virus and no cell-free virions are found inside an infected bird. Thus, the CTL response is believed by the field to be the primary mechanism needed to kill infected cells and provide anti-tumor effects. Put another way, antibodies have shown limited effect on both viral levels and disease incidence.

5. Check the references used in this study.

All the references have been checked.

Reviewer 2 Report

General Comment: 

In this manuscript, the authors investigated the role of CD8 T cells in vaccine protection against MDV. They successfully produced chicken CD8-specific antibodies and used them to deplete CD8 T cells in vaccinated and unvaccinated animals. They could demonstrate that already a reduction of CD8 T cells by 2-3 fold dramatically increases disease and tumor incidence. The study is well written and contains important information on the immune response needed to protect against MDV. The study contains very important information to the field and beyond, and is publishable after minor revisions.

Minor Points:

  • Line 19-22: these two sentences are redundant. Please combine them as they basically state the same.
  • Figure 1: the resolution of parts of the figure is suboptimal and should be improved. Personally, I like the cartoon style of the chicken and syringes, but not every reader likely feels the same way about it.
  • Figure 2: the resolution should be improve, especially of figure panel 2C-G. In addition, labels should be included in the figure for panel C, D, E and F (control, CD8 antibody, spleen and thymus), which would make this figure more intuitive to understand. Right now, it is very tedious to look up the description in the figure legend.
  • Line 63: the authors cannot claim that “CD8+ T cells are essential for MD protection” based on their data. Replace “essential” by “important”.
  • Line 363f: This should be CVI988 not CVI1988
  • The authors used all vaccine generations including a bivalent vaccine, but did not test the gold standard vaccine CVI988. I’m sure they have a great rational for it that would be worth including in the discussion section.

Author Response

We greatly appreciate the prompt and thorough comments that we made by both reviewers. We have addressed each comment which is highlighted in yellow, as well as made changes in the manuscript that are also highlighted in yellow. We hope that these modifications make the manuscript more valuable to the community.

Reviewer #2: In this manuscript, the authors investigated the role of CD8 T cells in vaccine protection against MDV. They successfully produced chicken CD8-specific antibodies and used them to deplete CD8 T cells in vaccinated and unvaccinated animals. They could demonstrate that already a reduction of CD8 T cells by 2-3 fold dramatically increases disease and tumor incidence. The study is well written and contains important information on the immune response needed to protect against MDV. The study contains very important information to the field and beyond, and is publishable after minor revisions.

Minor Points:
Lines 19-22: these two sentences are redundant. Please combine them as they basically state the same.

The two sentences were combined as suggested.

Figure 1: the resolution of parts of the figure is suboptimal and should be improved. Personally, I like the cartoon style of the chicken and syringes, but not every reader likely feels the same way about it.

Figure 1 was revised as requested

Figure 2: the resolution should be improve, especially of figure panel 2C-G. In addition, labels should be included in the figure for panel C, D, E and F (control, CD8 antibody, spleen and thymus), which would make this figure more intuitive to understand. Right now, it is very tedious to look up the description in the figure legend.

Figure 2 was revised as requested by increasing the resolution of the images and adding descriptions in the figure.

Line 63: the authors cannot claim that “CD8+ T cells are essential for MD protection” based on their data. Replace “essential” by “important”.

Good point. The wording has been changed as suggested; see line 61.

Line 363: This should be CVI988 not CVI1988. The authors used all vaccine generations including a bivalent vaccine, but did not test the gold standard vaccine CVI988. I’m sure they have a great rational for it that would be worth including in the discussion section.

Thank you picking up the error, which has now been corrected; see line 382. As for not including CVI988, this information is included in the Discussion; see lines 383-390.